

# Tropane alkaloids and terpenes synthase genes of *Datura stramonium* (Solanaceae)

Sabina Velázquez-Márquez, Iván M. De-la-Cruz, Rosalinda Tapia-López and Juan Núñez-Farfán

Laboratorio de Genética Ecológica y Evolución, Departamento de Ecología Evolutiva, Instituto de Ecología, Universidad Nacional Autónoma de México, Ciudad de México, DF, Mexico

Corresponding author
Juan Núñez-Farfán, farfan@unam.mx

## ABSTRACT

**Background**. Plants have evolved physical–chemical defense to prevent/diminish damage by their enemies. Chemical defense involves the synthesis' pathways of specialized toxic, repellent, or anti-nutritive metabolites to herbivores. Molecular evolutionary studies have revealed the origin of new genes, acquisition and functional diversification along time in different plant lineages.

**Methods**. Using bioinformatic tools we analyze gene divergence of tropane alkaloids (TAs) and terpene synthases (TPSs) in *Datura stramonium* and other species of Solanaceae; compared gene and amino acids sequence of TAs and TPSs on genomes, cDNA and proteins sequences of Viridiplantae. We analyzed two recently assembled genomes of *D. stramonium* (Ticumán and Teotihuacán), transcriptomes of *Datura metel* and genomes of other Solanaceae. Hence, we analyzed variation of TAs and TPSs to infer genes involved in plant defense and plant responses before stress. We analyzed protein modeling and molecular docking to predict interactions between H6H and ligand; we translated the sequences (Teo19488, Tic8550 and Tic8549) obtained from the two genomes of *D. stramonium* by using Swiss-Model and Ramachandran plot and MolProbity structure validation of Teo19488 protein model.

**Results**. For TAs, we detected an expansion event in the tropinone reductase II (TRII) and the ratio synonymous/non-synonymous substitutions indicate positive selection. In contrast, a contraction event and negative selection was detected in tropinone reductase I (TRI). In Hy-oscyamine 6 b-hydroxylase (H6H), enzyme involved in the production of tropane alkaloids atropine and scopolamine, the synonymous/non-synonymous substitution ratio in its dominion indicates positive selection. For terpenes (TPS), we found 18 DsTPS in *D. stramomiun* and seven in *D. metel*; evolutionary analyses detected positive selection in TPS10.1 and TPS10.2 of *D. stramonium* and *D. metel*. Comparison of copies of TPSs in *D. stramonium* detected variation among them in the binding site. Duplication events and differentiation of TAs and TPSs of *D. stramonium*, as compared to other Solanaceae, suggest their possible involvement on adaptive evolution of defense to herbivores. Protein modeling and docking show that the three protein structures obtained of DsH6H from Teo19488, Tic-8550 and Tic8549 maintain the same interactions and the union site of 2OG-FeII_Oxy with the Hy-o ligand as in 6TTM of *D. metel*.

**Conclusion**. Our results indicate differences in the number of gene copies involved in the synthesis of tropane alkaloids, between the genomes of *D. stramonium* from two Mexican populations. More copies of genes related to the synthesis of tropane alkaloids

(TRI, TRII, H6H, PMT) are found in *D. stramonium* as compared to Viridiplantae. Likewise, for terpene synthases (TPS), TPS-10 is duplicated in *D. stramonium* and *D. metel*. Further studies should be directed to experimentally assess gain (overexpression) or loss (silencing) of function of duplicated genes.

## INTRODUCTION

Plants are exposed to manifold environmental factors, biotic and abiotic, that affect their lifetime reproductive success. To cope with the different types of environmental stress, plants have evolved different mechanisms, ranging from morpho-anatomical, physiological, biochemical, molecular and epigenetic modifications, among others (*Lamalakshmi et al., 2017*). An ubiquitous defensive mechanism that protects plants from physical and biotic stresses is represented by the synthesis of specialized metabolites (SM; the so-called "secondary metabolites"). Thousands of chemical compounds found in plants promote protection against plants' natural enemies (i.e., pathogens, viruses and herbivores) (*Mithöfer & Boland, 2008*). More than 40,000 and 12,000 terpenoids and alkaloids, respectively, have been described in plants (*Zhou & Pichersky, 2020*).

Tropane alkaloids (TAs) and Terpene synthases (TPSs) are important specialized molecules of plants that help to protect them from herbivores and pathogens (*Kessler & Baldwin, 2002*). Tropane alkaloids are distinctive, but not exclusive, of Solanaceae (*Wink, 2003*; *Mithöfer & Boland, 2008*; *Pigatto et al., 2015*) and in *Datura stramonium*, implicated in plant defense to herbivores (*Shonle & Bergelson, 2000*; *Castillo et al., 2014*; *Miranda-Pérez et al., 2016*). Terpene synthases are enzymatic genes involved in the synthesis of volatile organic compounds (VOCs) in many organisms (*Picazo-Aragonés, Terrab & Balao, 2020*). Alkaloids and VOCs negatively affect herbivores. The concentration of VOCs may even rise before herbivores start to feed (*Heil, Lion & Boland, 2008*). Recent molecular studies in plants indicate modifications in the expression of SM as defense mechanisms (*Mithöfer & Boland, 2008*). Such modifications may result from the composition of the herbivore community, their abundance and dominance (*Koerner et al., 2018*). In this study we present a bioinformatic analysis of the changes in genes involved in the synthesis of tropane alkaloids and terpene synthases in Solanaceae, with particular attention to *Datura stramonium*. Gene modifications may include changes in the regulatory sequence or in the dominion, in the length of genes (including duplication), and post-transduction modifications (*Defoort, Van de Peer & Carretero-Paule, 2019*). Here, we analyze events of duplication in Solanaceae, with reference to whether duplicates are involved in modifications of biosynthetic pathways of TAs and TPSs.

Generally, copies of genes can have different fates. If different copies are retained, they may undergo sub-functionalization, dividing the original function. Alternatively, one copy may undergo neofunctionalization, acquiring a new function. However, the most frequent

outcome is pseudogenization, whereby one copy becomes non-functional (pseudogene) and tends to disappear due to accumulation of deleterious mutations. A pseudogene shows strong similarity to the parental copy, which encodes a particular protein, but has one or more alterations such as premature stop codons, mutations that cause phase changes, deletions and/or insertions that prevent a protein from being functional (*Ohno, 1970*). It has been reported that in some species the retention of pseudogenes for a long time only accumulates neutral mutations. However, several questions arise regarding the survival mode of these inactivated genes (*Ohno, 1970*; *Lynch & Conery, 2003*; *Conan & Wolfe, 2008*; Wang et al., 2012). Some retained functions maintain developmental homeostasis, for instance, the ability to rapid respond to a wide variety of environmental cues (Huot et al., 2014). One first defense response of plants involves the activation of signaling mechanisms (*De-la-Cruz, Velázquez-Márquez & Núñez Farfán, 2020*), and therefore protein turnover. It is estimated that more than 80% of the proteins are degraded through the proteosome pathway (*Maltsev et al., 2005*). In recent years it has been reported that the responses of specialized metabolites are grouped into several multigenic families. Therefore, we aim to elucidate the evolutionary process of two gene families, TAs (Tropane alkaloids) and TPSs (terpenes synthases) (*Karunanithi & Zerbe, 2019*; *De-la-Cruz et al., 2021*). Studies conducted in Solanaceae have found rapid responses to biotic and abiotic stimuli, in a matter of minutes, of genes involved in the synthesis of specialized metabolites (atropine, scopolamine, VOCs) (*Brille, Loreto & Baccelli, 2019*). To date, several groups have reported different TAs and TPSs and have found similar results (*De-la-Cruz, Velázquez-Márquez & Núñez Farfán, 2020*).

In this study we analyze the evolutionary history of genes involved in the synthesis of tropane alkaloids and terpene synthase in the Solanaceae family and specifically in draft genomes of two plants of *Datura stramonium* from populations of México (Ticumán and Teotihuacán). Overall, our bioinformatic analyses infer duplication of the genes of the main biosynthetic pathways leading to specialized metabolites.

# MATERIALS AND METHODS

## Selection and bioinformatic analyses: identification of TAs and TPs, genomics, transcripts and protein sequences

We obtained the sequences of nucleotide, transcripts and proteins of TAs and TPSs. Nine genomes (Table 1) of Solanaceae species were sourced for protein coding genes and CDS (Coding Sequence) genomes from the Sol Genomics Network (see links in Table 1): *Nicotiana tabacum* (*Edwards et al., 2017*), *Nicotiana sylvestris* (*Sierro et al., 2013*), *Nicotiana attenuata* (*Xu et al., 2017*), *Nicotiana tomentosiformis* (*Sierro et al., 2013*), *Solanum pimpinellifolium* (*Razali et al., 2018*), *Solanum lycopersicum* (*The Tomato Genome Consortium, 2012*), *Solanum pennellii* (*Bolger et al., 2014*), *Solanum tuberosum* (*The Potato Genome Sequencing Consortium, 2011*), *Capsicum annuum*, CM334 v1.55 (*Kim et al., 2014*). In addition, data of *Datura stramonium* were extracted from protein whereas TAs and TPSs' sequence were obtained from the genome of *D. stramonium* from the Mexican populations of Ticumán, Morelos (18°44′28.19″N,
**Table 1  Genome of Solanaceae employed in this study.**

| Specie | Link | Link |
|---|---|---|
| *Nicotiana tabacum* | https://solgenomics.net/organism/Nicotiana_tabacum/genome | https://solgenomics.net/tools/blast/ |
| *Nicotiana sylvestris* | https://solgenomics.net/organism/Nicotiana_sylvestris/genome | https://solgenomics.net/tools/blast/ |
| *Nicotiana attenuata* | https://solgenomics.net/organism/Nicotiana_attenuata/genome | https://solgenomics.net/tools/blast/ |
| *Nicotiana tomentosiformis* | https://solgenomics.net/organism/Nicotiana_tomentosiformis/genome | https://solgenomics.net/tools/blast/ |
| *Solanum pimpinellifolium* | https://solgenomics.net/organism/Solanum_pimpinellifolium/genome | https://solgenomics.net/tools/blast/ |
| *Solanum lycopersicum* | https://solgenomics.net/organism/Solanum_lycopersicum/genome | https://solgenomics.net/tools/blast/ |
| *Solanum pennellii* | https://solgenomics.net/organism/Solanum_pinnelii/genome | https://solgenomics.net/tools/blast/ |
| *Solanum tuberosum* | https://solgenomics.net/organism/Solanum_tuberosum/genome | https://solgenomics.net/tools/blast/ |
| *Capsicum annuum* | https://solgenomics.net/organism/Capsicum_annuum/genome | https://solgenomics.net/tools/blast/ |

99°7′44.26″W), and Teotihuacán, Estado de México (19°41′16.58″N, 98°50′4.14″W) (https://github.com/icruz1989/Datura-stramonium-genome-project) (*De-la-Cruz et al., 2021*) and transcriptomes (https://medplantrnaseq.org/).

We downloaded the whole genome of *D. stramonium* to extract the TPSs genes and gene prediction was performed using AUGUSTUS (*Stanke et al., 2006*). All sequences were uploaded and mapped using as reference species of Solanaceae. A total of 366,000 genes were extracted, and gene redundancy was reduced with CDHIT (*Chen et al., 2016*). The selected TPSs genes were blasted against the NCBI, UNIPROT databases and the nine genomes of Solanaceae, using a threshold value of $1E^{-5}$. All TPSs were downloaded from Viridiplantae followed by a protein-protein blast; hits above 80% of similarity and 70% of cover were saved for further analyses. Meta-alignments of protein sequences were performed using T-coffee (*Wallace et al., 2006*), MUSCLE v3.8 (*Edgar, 2004*) and MAFF (https://mafft.cbrc.jp/alignment/server/) v6. (*Rozewick et al., 2019*).

## Identification of orthogroups and inference of phylogenetic tree of TPa and TPSs

Construction of orthogroups, gene families, was performed according to *De-la-Cruz et al. (2021)* using Orthofinder v2.3.3 (*Emms & Kelly, 2015*; *Emms & Kelly, 2019*). The phylogenetic tree inference was accomplished using the program BEAST with the pair-wise deletion option, and reliability of the obtained phylogenetic tree was tested by bootstrapping with 1,000,000 MKKM chains. The model of substitution selected was JTT+G+F, with a branch support of 1 (*Drummond et al., 2002*). The trees were edited at the interactive tree of life (https://itol.embl.de) (*Letunic & Bork, 2019*).

We downloaded all orthologous sequences of TAs and TPSs of *D. stramonium*, *D. metel* and nine species of Solanaceae (see Table 1) to construct the orthogroups. Downloaded sequences of proteins and ORFs were extracted using TransDecoder and corroborated with GenScan. TPS genes were translated with GenScan and checked for the right direction and correct CDS; we corroborated this with the NCBI. The database used for BLAST was Uniprot; all available data for the analyzed plant species were downloaded. These amounts 847,544 sequences of TAs and TPSs. BLAST was carried out with a filtering E-value $< 1e^{-5}$ and only hits above 85% were selected (Data S1, S2). Orthogroups were

inferred using Orthfinder v 2.3.3. DIAMONS (*Emms & Kelly, 2015*; *Emms & Kelly, 2019*) while OrthoMCL was used to assess orthologs among species (*Chen et al., 2006*).

## Protein modeling

Molecular modelling and protein-ligand docking was performed with GLIDE software (*Schrödinger Release 2021-1, 2021*). Simulations of interactions for each protein was done in a quadrat of 6LU/ processed with the same coordinates of the crystalized ligand as described in *Wallace & Zhang (2020)*.

The SWISS-MODEL (http://www.expasy.org/swissmod/SWISS-MODEL.html) (*Waterhouse et al., 2018*) was employed to model the tertiary structure of proteins, from the previously determined structure by the dominion DIOX_N (PF14226) (*Hagel & Facchini, 2010*) as a blueprint of modelling. Models were visualized and analyzed through Visual Molecular Dynamics (*Waterhouse et al., 2018*; *Bienert et al., 2017*)

## Molecular docking

The in-silico docking (AutodockVina) of the PDB files (produced by the SWISS-Model) of Tic8550 (Ticumán) and Teo19488 (Teotihuacán) was performed in a simulation of molecular coupling with the native ligand of H6H, which is Hy-oscyamine (Hy-o). For docking, the protein was prepared with AutodockTools, assigning the parameters of energies and force fields of AutoDock4.0. Then, we obtained the results of the thermodynamically favored conformation for the Hy-o ligand using the algorithm VINA. The selected conformation had affinity coupling parameters of −7.3 and −7.5 kcl/mol for Teo19488 and Tic8550, respectively. As positive control, we performed the re-docking of the molecule of reference 6TTM (*Datura metel*), eliminating the ligand of the crystallized structure; the adopted conformation is similar to that obtained by the method of crystallization, and the coupling affinity for the reference model was −7.9 kcal/mol. The produced models are theoretically correct, and it is predicted that these have an oxidoreductase and use Hy-o as substrate.

### *Tropane alkaloids*

Analyses detected variation in the number of TAs gene copies in *D. stramonium*: DsTRI, DsTRII, DsPMT and DsH6H. Compared to Viridiplantae *D. stramonium* has one or more copies of these genes suggesting a genome expansion of TAs.

We detected eight copies of tropinone reductase I (DsTRI) and four copies of DsTRII. TRI and TRII define two groups in Solanaceae (Fig. 1A). The gene DsTRII have had duplications events in *D. stramonium* (Fig. 1B). The first duplication occurs in Rosidae-Asteridae, followed by three duplications at the base of Solanaceae and one more in *D. stramonium* (Fig. 1B). The phylogeny and structure of 32 protein sequences (TRI) from different plant species show variability in the ADH_Short_C2 TRI dominion (Fig. 2); in *D. stramonium* (Tic23_dati33027) there is a duplication of this dominion but shorter (Fig. 2).

In Hy-oscyamine- 6-ß-hydroxylase (*DsH6H*) in the Ticumán genome we detected two copies of the gene, Tic8550 and Tic8549 (Fig. 3A). Tic8550 has a tandem duplication of the DIOX-N dominion of 79 amino acids (Fig. 3) whereas Tic8549 has the dominion 2-oxaglutarate (65 amino acids) (Fig. 3A). In contrast, in the Teotihuacán genome, *DsH6H*
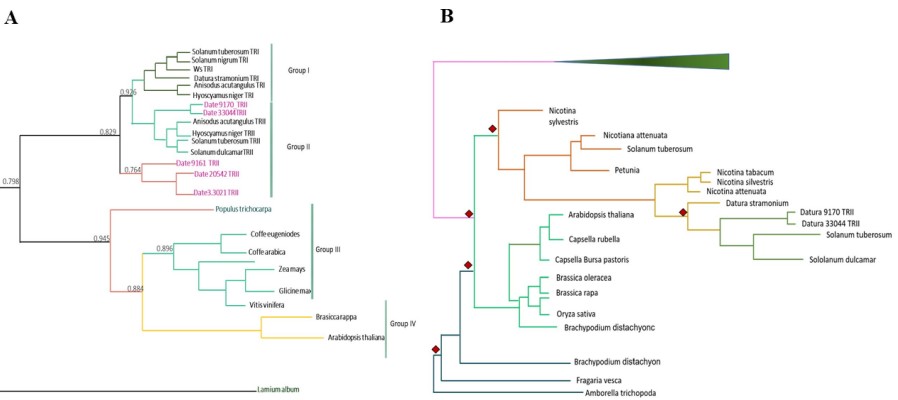

**Figure 1** **Phylogenetic analysis of TRI and TRII protein sequences.** TR-I and TR-II define Group I and II, respectively, in Solanaceae (A). Substitution rates ($\omega$) of *TR-II* (B). Three duplication events in the genome of *Datura stramonium* are indicated (diamonds).

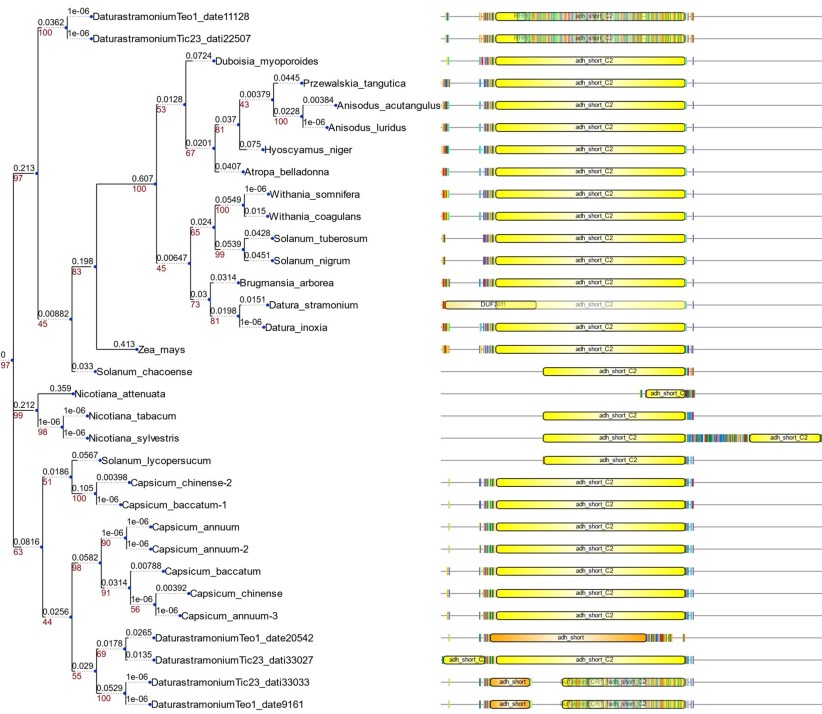

**Figure 2** **Structure of domains of TRI.** Phylogenetic analysis and structure of domains of TRI gene in different species. Six copies are distributed in different clades. Tic23dati33027 has an extra domain adh_short_C.

(i.e., Teo19488) has only one copy (Fig. 3). The alignment of 29 sequences of H6H of different plant species indicate the variation in the duplicated DIOX-N dominion in Tic8550 (Fig. 3B).

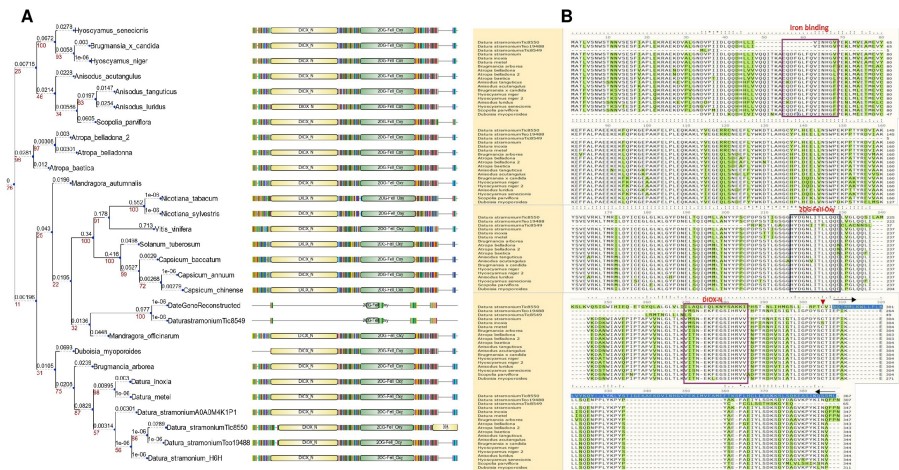

**Figure 3 Phylogeny of gene H6H.** Phylogeny of gene H6H in 19 species. In (A) we observe two copies of the gene are present in the two sequenced genomes of *D. stramonium* (Tic and Teo). In Tic8550 there is a tandem duplication of dominion DIOX N (PF14226) vs. Teo 19488; two copies of the gene are present in both genomes (segmental duplication) distributed in different clades. (B) Alignment of sequences of protein H6H with the conserved zones highlighted. In the terminal carboxyl, the DIOX-N dominion is duplicated in TIC8550.

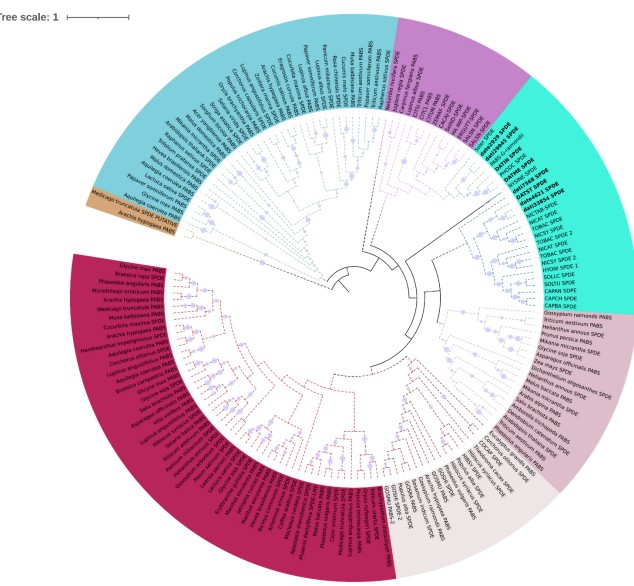

**Figure 4 Phylogeny of pmt.** Phylogeny of Putrescine N-methyltransferase (*pmt*). date = *Datura stramonium* Teotihuacán, dati = *Datura stramonium* Ticumán. The genomic analyses indicate an expansion of *pmt*, two copies in date and five in dati. The latter has an additional dominion of spermine synthase.

In N-methyl putremescine transferase (DsPMT) we found variation in the number of gene copies in comparison to other Solanaceae (Fig. 4). In Tucumán's genome there are three copies while two copies were detected in Teotihuacán's genome (Fig. 4, Fig. S1).

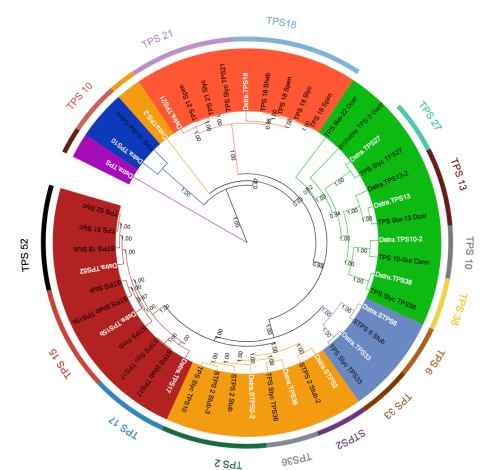

**Figure 5   Phylogeny of TPS.** 18 TPSs were found in *D. stramonium* (white letters) and Solanaceae and other angiosperms. Two copies of TPS-10 and TPS-13 in the subfamily TPSa. Tree obtained by Bayesian inference, with JTT+G+F evolutionary model. Most branches have a bootstrap support of 1.

### Terpene synthases

Eighteen TPSs genes were found in the genomes of *D. stramonium* (Fig. 5; Table S1) and seven in *Datura metel* (Fig. S2). The 18 *DsTPS* found in *D. stramonium* are distributed in four subfamilies, identified for other angiosperms: Nine in TPSa, 3 in TPSb, 2 in TPSc, 2 in TPSg and 2 TPS- unknown (Fig. S3, Tables S1 and S2) (*Huang et al., 2013*; *Huang et al., 2017*; *Falara et al., 2011*).

The domains of these TPSs, directly involved in the biosynthesis of terpenoids, show expansion events and positive selection in TPS10 (Fig. 6; Table S3). Changes in the dominions are: Terpene synthase, N-terminal domain (IPR001906), Terpenoid cyclases/protein prenyl transferase alpha-alpha toroid (IPR008930), Terpene synthase, metal-binding domain (IPR005630), Terpene cyclase-like 1, C-terminal (IPR034741). Analyses indicate that Solanaceae have the gene TPS10.1 whereas only *D. stramonium* and *D. metel* possess TPS10.2 (Fig. 7 and Tables S2 and S3); this possibly is a duplicate of TPS10.1_like.

## Protein modeling and molecular docking

Superimposition of the reported protein structure of H6H from *D. metel* (PDB ID 6TTM) with the corresponding model obtained from *DsH6H* sequences for the two genomes of *D. stramonium* from two Mexican populations (Teo19488 and Tic8550), indicate similarities (Fig. 7A). The Teo19488 predicted structure match very well with the DmH6H structure (Fig. S2); similarly, Tic8550 matches well except for some residues that were not modeled (Fig. 7B, Fig. S3). This fragment of Tic8550 have a big terminal protein sequence (NQTMMLKWLLFLKLVYKQYLYKLYSTIGEKAEKDVINHGVPEKIMVEAMEFTKSFL-HCLLRKKKSLSQKEASIKQSSML) of 79 amino acids was not modeled (Fig. 7B, cf. Fig. 3B). Further, a NCBI protein-protein blast of this sequence aligns well to the region: 33–72 of DmH6H fragment; we analyzed this fragment separately as Tic8550_Fragment (Fig.

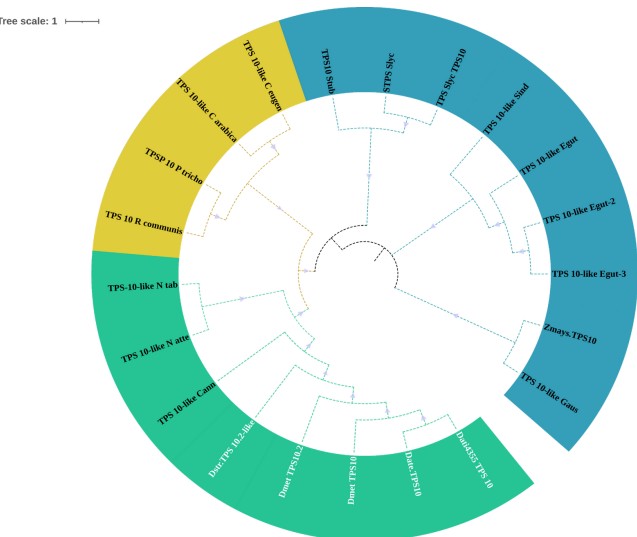

**Figure 6 Phylogeny of TPS-10.** Phylogeny of TPS-10 (green) of *Datura stramonium*, *D. metel* and other Solanaceae. In these species the TPS-10 is duplicated. Tree obtained by Bayesian inference, with JTT+G+F evolutionary model. Most branches have a support above 80%.

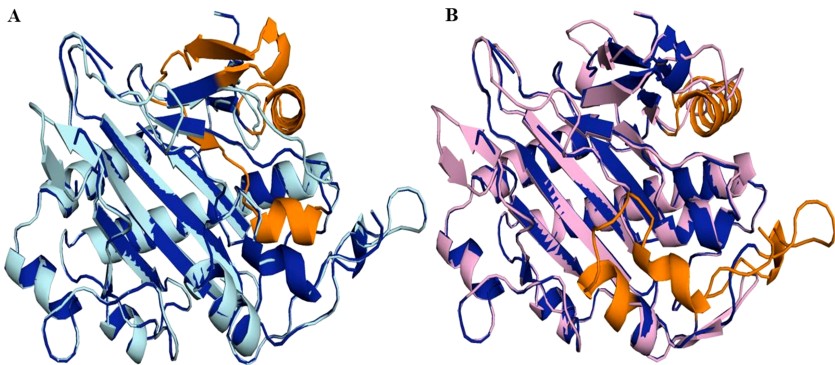

**Figure 7 Structure of 6TTM in *Datura metel*.** (A) Superimposition on the tertiary structure of H6H reported for *Datura metel* (PDB ID 6TTM) (in blue) of obtained models of the sequence for gene DsH6H *Datura stramonium* Teo19488 (Cyan) (A) and Tic8550 (Pink) (B). In orange color the folding present only in the reference structure 6TTM but not in *D. stramonium* are highlighted.

86). The Tic8549 sequence corresponds to a 65 amino acids sequence (MLPIIPRPKSTL-GAGGHYDGNIITFLQQDCLACNNSLLRMTNGLLLNLSYCFCGLSGTHSKGYEQ) that was modeled as Hy-oscyamine 6 beta-hydroxylase like fragment (Fig. 8). This fragment (Tic8549) aligns to DmH6H in one section of the binding pocket, where His-217 and Asp-219 are coordinated with $Ni^{2+}$ (Fig. 8C). The presence of these amino acids in Tic8549 sequence is interesting since $Ni^{2+}$ ion, a surrogate of the natively present $Fe^{2+}$ ion, is coordinated by the side chains of His-217 (strand $\beta$II), Asp-219 (loop $\beta$II/ $\beta$III) and His-274 (strand $\beta$VII) that forms a metal binding **His-X-Asp...His** motif, highly conserved in the oxoglutarate dependent oxygenases (ODD) family (*Kluza et al., 2020*).
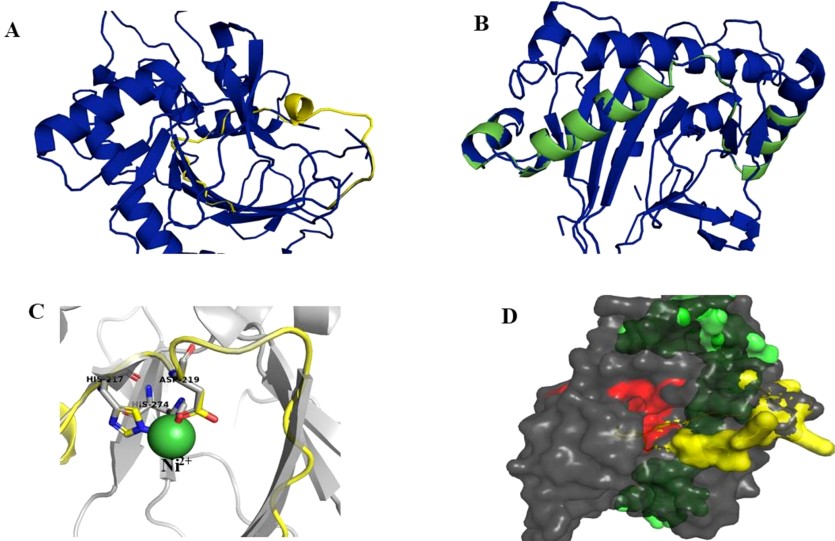

**Figure 8  Models of H6H.** Superimposition of reported structure of DmH6H (PDB ID 6TTM) (blue) and models obtained for the sequence Tic8549 (yellow) (A) and a fragment of the sequence Tic8550 (green) (B). Both sequences possess structural similarity with other sequences of the same protein. The fragment Tic8549 (yellow) aligns in the region where residues of the active site of H6H are located (gray) and involved in the coordination of the structure with metals (C). This site is highly conserved in the enzyme families ODD. (D) surface of fragments Tic8549 (Yellow) and Tic8550_Fragment (green) superimposed on H6H structure. In red are the residues that compose the active (binding) site.

Protein models of H6H for *D. metel* and *D. stramonium* (Teo19488 and Tic8550) predict more interaction between residues in Tic8550 than on *D. metel* and Teo19488 (Fig. 9). It has recently been reported that the binding pocket (BP) of DmH6H is mainly formed by hydrophobic amino acids. It has been demonstrated that in the crystalized H6H the phenyl ring of Hy-o is bounded in an aromatic cage formed by Phe-103, Tyr-295, Tyr-319, Phe-322 and Tyr-326, of which the most prominent bound is with Tyr-326 which forms CH-$\pi$ hydrogen bonds with the phenyl ring of Hy-o in an edge-to-face bidentate manner (*Kluza et al., 2020*) (Figs. 9A, 9D). Our results shown that the predicted model for Teo19488 have the same amino acids forming the aromatic cage composed by Phe-88, Tyr-255, Tyr-279, Phe-282 and Tyr-286 and the predicted pose of Hy-o obtained by the docking in silico shows that the phenyl ring is inside of this aromatic cage (Fig. 9B). Nevertheless, the main interaction with Tyr-286 was not detected (Fig. 9E). This could be because the cutoff distance was set 4 Å and the predicted pose of this residue is beyond this cutoff distance. The predicted structure of Tic8550 has different amino acids arrangement inside the binding pocket. This predicted structure does not contain the aromatic cage, instead it is predicted that the phenyl ring of Hy-o inside of a cavity formed by the non-polar amino acids Leu-308, Leu-92, Ile-179, Met-181, Met-292 and the polar uncharged amino acids Asn-206 and Asn-289 (Fig. 9C). Also, it is predicted that the phenyl ring has several non-polar interactions with these side chain amino acids (Fig. 9F). The Asn-206 forms a

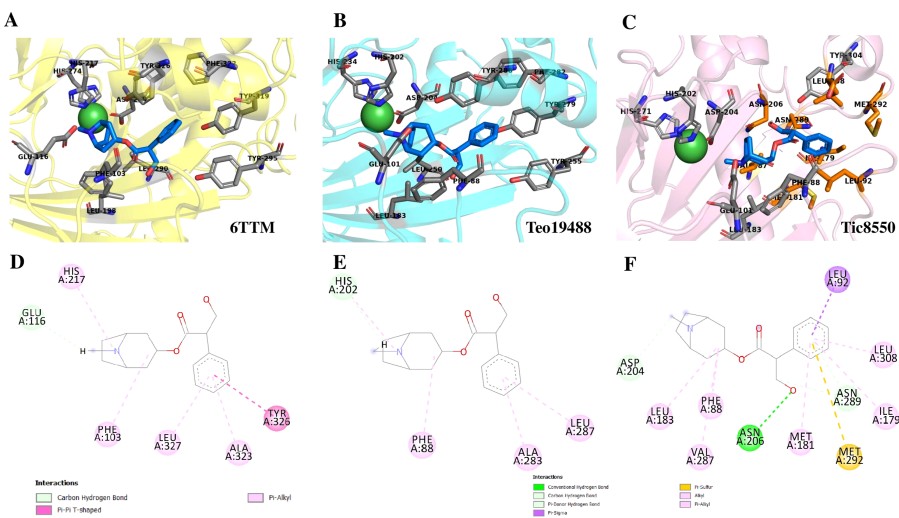

**Figure 9 Structure of 6TTM.** (A) Structure of 6TTM in *Datura metel* (yellow). The ligand (blue) inside the BP is surrounded by amino acids (gray). (B) Structure modelled for *Datura stramonium* Teo_19488 (cyan) and (C) *D. stramonium* Tic_8550. The interactions between the predicted residues of the BP and the ligand Hy-o are illustrated in D, E and F.

conventional hydrogen bond with the hydroxyl group from Hy-o which may stabilize the structure in the BP.

## DISCUSSION

Gene duplicates product of molecular evolution are the raw material for evolutionary innovation (*Defoort, Van de Peer & Carretero-Paule, 2019*). Recent findings have uncovered the interactions and contrasting functions of genes that may help plants to confront environmental stresses in nature and serve as important agronomic characteristics. Hence, to trace the origin, molecular mechanisms, evolutionary fate and function of gene duplicates is a main goal (*Panchy, Lehti-Shiu & Shiu, 2016*; *Soltis & Soltis, 2016*; *Van de Peer, Mizrachi & Marchal, 2017*). The phylogenetic analyses of genes involved in the synthesis of tropane alkaloids (*DsPMT, DsTRI, DsTRII* and *DsH6H*) here presented, indicate that two genes (DsTRII and DsH6H) have expanded in *D. stramonium*, presenting five and two copies, respectively. Previous studies have shown that tropinone reductases are involved in the bifurcation of the synthesis pathways in Solanaceae (*Dräger et al., 1992*; *Nakajima, Hashimoto & Yamada, 1993*; *Birgit, 2006*; *Kanehisa & Sato, 2019*). These genes are responsive to environmental stimuli and activate the signaling pathways (*Campos, Kang & Howe, 2014*). The presence of these genes in the Solanaceae, as revealed by phylogenetic analyses, suggests that these duplicates in *D. stramonium* may contribute to increase the production of tropane alkaloids. The absence of TRII in *Zea mays* and *A. thaliana* suggest that these species never had this gene or lost it during their evolution. However, the gene is also absent in species of other clades (e.g., *Brassica rapa, Glycine max, Vitis vinifera,* etc.; Fig. 1). Another possibility is that TRII is not related to the synthesis of tropane alkaloids in these species (*Miller, Arteca & Pell, 1999*).
The gene DsPMT of *D. stramonium* found in the population of Ticumán, México, had an extra domain of spermine-synthase in comparison with its homologous gene found in the same species but from the population of Teotihuacán, México (*De-la-Cruz et al., 2021*). We named this gene as *Ds* PMT7568ti (Fig. 4), Spermine gene is a potent plant defense activator, with protective effects of broad-spectrum (*Seifi & Shelp, 2019*). Reports indicate that overexpression of spermidine synthase enhances tolerance to multiple environmental stresses, including the attack by herbivores and pathogens (*Kasukabe et al., 2004*; *Seifi et al., 2019*). In addition, PMT is a key enzyme in the catalysis of N-methylputrescine from putrescine and S-adenosyl-L-methionine and triggers the production of hygrine and other different tropane alkaloids (*Kanehisa & Sato, 2019*).

In contrast to other plants, in Solanaceae (*Atropa belladona*, *Hy-osciamus niger* and *Datura stramonium*), TRII is duplicated and under positive selection. In this line, we hypothesize that TRII may have acquired a different functional importance after its duplication in *D. stramonium* and hence, it has retained multiple copies of the gene. Yet, this hypothesis warrants further analysis. The phylogeny indicates an independent history, for instance, from Brassicaceae.

Perhaps TRs in Brassicales have other functions, different to the production of alkaloids, but using the same pathways. For instance, they may have conferred plasticity to deal with environmental stresses. Plant species with small genome size may have lost considerable fractions of their genome. Thus, enzymes with similar function may have been replaced by TRII.

Of the four subfamilies of TPSs found in the *D. stramonium*'s genome, three have been reported in other angiosperms (TPSa, TPSb and TPSg). TPS10 is member of the subfamily *a* and its role in defense against pathogens and herbivores has been dilucidated in *Zea mays* (*Huang et al., 2017*; *Singh & Sharma, 2015*; *Tholl, 2006*). We detected positive selection on TPS; positive selection on TPS10.1 and TPS10.2 was detected in *D. stramonium* and *D. metel.* We speculate that the presence of two copies of this gene can contribute to plant defense against plants' natural enemies (*Köllner, Gershenzon & Degenhardt, 2009*; *Köllner, Degenhardt & Gershenzon, 2020*).

In this line, duplicated genes involved in the specialized metabolism and plant defense often show differential restrictions (constraints, trade-offs, limitations) either indicating positive selection or differential selection (*Wang et al., 2016*). Apparently, TRII displays genic redundancy since it possesses different copies and polymorphic sites. However, this may not be the case since this process is only present in genes of primary metabolism or development (*Wang et al., 2016*; *Maltsev et al., 2005*) which are constitutive and display low variation. The gene H6H is involved in the last step in the production of scopolamine and atropine in one of the pathways for the production of tropane alkaloids (*Kanehisa & Sato, 2019*). Here, we detected one duplicated in tandem of this gene in *D. stramonium* (Ticumán has two domains of DIOX_N (PF14226), although each copy belongs to two different gene families (OG0028637 and OG0043057; cf. Fig. 3). Similarly, we found 18 TPSs in *D. stramonium* that likely are involved in plant defense against herbivores like in other plants species (e.g., Bharat & Sharma 2015). Lineages in the Solanaceae have high values of the ratio $K$a/$K$s (Table 2). Thus, positive selection may have acted on TPS10.1

**Table 2  Selection test for tropane alkaloids (TAs) and terpene syntahses (TPSs) genes of *Datura stramonium*.**

| Genes | Chi-Square | Proportion of sites selected | $\omega$ |
|-------|-----------|------------------------------|----------|
| *PMT* | 5.43432 | 0.1389 | 2.5778 |
| *TRI* | 0 | NA | |
| *TRII* | 31.5335 | 0.09994 | 10.6356 |
| *H6H* | 15.66146 | 0.15136 | 6.469 |
| *TPS10_1* | 22.75614 | 0.07163 | 1.704 |
| *TPs10_2* | 16.16614 | 0.014589 | 2.295 |
| *TPS14* | 4.67532 | 0.098812 | 3.7809 |
| *TPS21* | 1.43981 | NA | |
| *GGPS1* | 3.89631 | 0.0899 | 2.5988 |

and TPS10.2 of *D. stramonium*, not recently and the selection effect on allelic diversity may be masked by the accumulation of neutral mutation.

On the other hand, results of protein simulation and molecular docking suggest that the structure and interactions of this protein (H6H) in Tic8550 might have a different structure to that of the known H6H; we speculate if this may be related to enhancing the efficiency of alkaloids' synthesis.

It is interesting to note that the Tic8549 possess only the fragment 2OG-FeII_Oxy (PF03171) whereas Tic8550 possess an extra dominion (DIOX_N) which aligns structurally to a highly conserved region of H6H proteins (Figs. S2 and S3) (and associated to a coordination of binding metal $Ni^{+2}$ (Fig. S5)). Further, it is interesting noting that a particular gene Tic8549 is associated to this fragment DIOX_N (PF14226).

Finally, the number of copies and differentiation of TAs and TPSs of *D. stramonium*, as compared to other solanaceous species, suggest their involvement on adaptive evolution of chemical defense against herbivores. Further studies should be directed to experimentally assess gain (overexpression) or loss (silencing) of function of duplicated genes.

## CONCLUSION

Our results indicate differences in the number of gene copies involved in the synthesis of tropane alkaloids, between the genomes of *Datura stramonium* from the two Mexican populations (Ticumán and Teotihuacán). Furthermore, more copies of genes related to the synthesis of tropane alkaloids (TRI, TRII, H6H, PMT-10) are found in *D. stramonium* as compared to Viridiplantae. Likewise, for terpene synthases (TPS), TPS-10 is duplicated in *D. stramonium* and *D. metel*. These results point future experimental studies of gain/loss of duplicated genes.

Links to data on Figshare (Data S3).

## ACKNOWLEDGEMENTS

We are grateful to the three reviewers whose comments and suggestions improved this contribution. We thank to Tonatiuh Campos García his advice in protein modelling. To

the members of the *Laboratory of Ecological Genetics and Evolution* for encouragement and help.

### Funding

This work was financed by a CONACyT grant "Genomics of plant defense against their natural enemies" (# 1527) and PAPIIT UNAM grant (#IG200717) granted to Juan Núñez-Farfán. Sabina Velázquez-Márquez was granted a Postdoctoral fellowship by CONACyT # 2019-000019-01NAC. Iván M. De-la-Cruz received a Doctoral scholarship from CONACyT # 283799. The funders had no role in study design, data collection and analysis, decision to publish, or preparation of the manuscript.

### Grant Disclosures

The following grant information was disclosed by the authors:
CONACyT grant Genomics of plant defense against their natural enemies: # 1527.
PAPIIT UNAM: #IG200717.
Postdoctoral fellowship by CONACyT: #2019-000019-01NAC.
Doctoral scholarship from CONACyT: # 283799.

### Competing Interests

The authors declare there are no competing interests.

### Author Contributions

- Sabina Velázquez-Márquez conceived and designed the experiments, performed the experiments, analyzed the data, prepared figures and/or tables, authored drafts of the paper, and approved the final draft.
- Ivan M. De-la-Cruz performed the experiments, analyzed the data, prepared figures and/or tables, and approved the final draft.
- Rosalinda Tapia-López performed the experiments, prepared figures and/or tables, and approved the final draft.
- Juan Núñez-Farfán conceived and designed the experiments, performed the experiments, co-authored drafts of the paper, and approved the final draft.

### DNA Deposition

The following information was supplied regarding the deposition of DNA sequences:

Whole genome of two Mexican plants (Teo and Tic; reported at *Scientific Reports* https://doi.org/10.1038/s41598-020-79194-1) of *Datura stramonium* is available at GitHub: https://github.com/icruz1989/Datura-stramonium-genome-project.

The genome assemblies, Illumina and PacBio raw sequences from the two plants of *D. stramonium*, are available at DDBJ/ENA/GenBank, BioProject PRJNA622882: Teotihuacan assembly, acc. JAAWWX000000000, Ticumán assembly acc. JAAWWY000000000.

The Illumina and PacBio sequences for the Ticumán genome are available at SRR11474700 and SRR11474698, respectively. The Illumina and PacBio sequences for the Teotihuacán genome are available at: SRR11474701 and SRR11474699, respectively.

The two transcriptomes of *Datura stramonium* (Datura_stramonium_1.tgz and Datura_stramonium_2.tgz) and one of *Datura metel* (Datura_metel.tgz) are available at: https://medplantrnaseq.org/.

## Data Availability

The TPS proteins of *Datura stramonium* and D. metel sequence are available in the Supplemental Files.

The data is available at:

Velázquez, Sabina; Fárfan, Juan Núñez (2021): TPS gene analysis. figshare. Dataset. https://doi.org/10.6084/m9.figshare.14343356.v3

Velázquez, Sabina; Fárfan, Juan Núñez (2021): TPS genome blast. figshare. Dataset. https://doi.org/10.6084/m9.figshare.14343347.v2

Velázquez, Sabina; Fárfan, Juan Núñez (2021): Augustus analysis. figshare. Dataset. https://doi.org/10.6084/m9.figshare.14343320.v2

Velázquez, Sabina; Fárfan, Juan Núñez; De la Cruz, Ivan; Tapia-López, Rosalinda (2021): *Datura metel* transcriptome analysis. figshare. Dataset. https://doi.org/10.6084/m9.figshare.14343317.v2

Velázquez, Sabina; Fárfan, Juan Núñez (2021): Transcriptome analysis D. stramonium and D. metel. figshare. Dataset. https://doi.org/10.6084/m9.figshare.14343302.v2

Velázquez, Sabina; Fárfan, Juan Núñez (2021): Transcriptome assembly D. metel. figshare. Dataset. https://doi.org/10.6084/m9.figshare.14343296.v2

Velázquez, Sabina; Fárfan, Juan Núñez (2021): TPS Genes and Tropane Alkaloids. figshare. Dataset. https://doi.org/10.6084/m9.figshare.14343284.v2

Velázquez, Sabina; Fárfan, Juan Núñez (2021): TPS10. figshare. Dataset. https://doi.org/10.6084/m9.figshare.14343278.v1

Velázquez, Sabina; Fárfan, Juan Núñez (2021): Tropinone reductase. figshare. Dataset. https://doi.org/10.6084/m9.figshare.14343272.v3

Velázquez, Sabina; Fárfan, Juan Núñez (2021): PMT Tropane alkaloid. figshare. Dataset. https://doi.org/10.6084/m9.figshare.14343269.v4

Velázquez, Sabina; Fárfan, Juan Núñez (2021): H6H Tropane alkaloid. figshare. Dataset. https://doi.org/10.6084/m9.figshare.14343263.v8.

## Supplemental Information

Supplemental information for this article can be found online at http://dx.doi.org/10.7717/peerj.11466#supplemental-information.

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
