# Peer review of "Tropane alkaloids and terpenes synthase genes of Datura stramonium (Solanaceae)"

_PeerJ, doi:10.7717/peerj.11466_

## Round 0.1 · original submission · Major Revisions

We have received three detailed reviews, that will be extremely helpful for the improvement of this manuscript. I recommend that authors prepare a point-by-point response to every comment made by the reviewers.

· Appeal

Appeal


· · Academic Editor

Reject

This decision is based on the recommendations and comments of reviewers.

The main reason is commented by reviewer #1:

"The study is far too preliminary to be considered for publication and interpretation of the data is flawed. The authors claim to be using sequences derived from a genome sequence of Datura stramonium that is not publically available and none of the sequences is provided in the manuscript or submitted to Genbank. Hence there is no possibility for anyone to independently confirm their findings. In addition, throughout this study, accession numbers or gene identifiers are not provided for any of the sequences used to construct the phylogenetic trees. It is impossible for the reader to understand the identity of the genes that are being compared. No raw data is shared. "

Reviewer 1 ·

Basic reporting

The data presented in this manuscript attempts to investigate the evolution of genes involved in tropane alkaloid and terpene biosynthesis in the Solanaceae family. The study is far too preliminary to be considered for publication and interpretation of the data is flawed. The authors claim to be using sequences derived from a genome sequence of Datura stramonium that is not publically available and none of the sequences are provided in the manuscript or submitted to Genbank. Hence there is no possibility for anyone to independently confirm their findings. In addition, throughout this study, accession numbers or gene identifiers are not provided for any of the sequences used to construct the phylogenetic trees. It is impossible for the reader to understand the identity of the genes that are being compared. No raw data is shared.

Furthermore, the study is focused on investigating gene duplications and losses and one of the claims states that TRII has expanded in the Solanaceae, particularly Datura stramonium, while TRI copy number has contracted. There is absolutely no evidence to support these data. TRI and TRII are both members of the short-chain dehydrogenase / reductase superfamily and plant genomes contain many sequences that are related to these genes. They encode enzymes that are catalytically promiscuous and act on many ketones to produce alcohols. The authors provide no biochemical evidence to indicate that the sequences they have identified actually encode enzymes with TRII activity. Hence the assumption that these enzymes contribute to tropane alkaloid biosynthesis is flawed. Indeed it is highly likely that these enzymes catalyze other reactions. Similarly, analysis of the terpene synthase sequences is also fundamentally flawed. Numerous other studies have utilized phylogenetic analyses to place terpene synthases into several subfamilies, which can be used to infer function. For example, tomato contains close to 50 terpene synthase genes whereas the current study presents five sequences from Datura and is therefore almost certainly incomplete. The phylogenetic framework used by the authors has not taken into account the sub-groupings adopted by the terpene field.

Finally, specialized metabolism is inherently complex. You cannot infer gene function based on phylogeny alone. The data presented by the authors represents a potential start of future studies to investigate their findings and the function of the genes they have identified. As presented, this data alone is not suitable for publication.

Experimental design

no comment

Validity of the findings

no comment

Reviewer 2 ·

Basic reporting

Topic of great interest and the article is original. Clear study written in professional English with sufficient bibliographic references. The article structure was conform. However, document may be better presented.

Experimental design

The study meets the requirements of the journal with clearly defined objectives. The scientific approach was rigorous.

Validity of the findings

Research results were clear and well discussed. They linked to the original research question.

Additional comments

Topic of great interest and the article is original. However, document may be better presented.

Abstract:
Specify, in the abstract, that the study is carried out using bioinformatics tools.

Main text:

All abbreviations should be clarified on first instance (for example: line 77, CDS);

Correct the title on line 115: Conserved protein motifs and exon/intron structure de TAs and TPSs genes;

Homogenize the references in the main text, for example, in the line 57: (Wang et al 2012), in the line 68: (Brille et al., 2019), in the line 80: (Xu et al. 2017) and in the line 100: (Mulder N.J et al., 2005);

Reference for line 91 dated 2022?! (Drummond, et al., 2022). Rectify.

·

Basic reporting

THIS STUDY COULD BE COMPLEMENTED WITH CHEMISTRY AS WELL. The authors are using sequences derived from a genome sequence of Datura stramonium that is not publically available and none of the sequences is provided in the manuscript or submitted to Genbank. Lack of accession numbers or gene identifiers to confirm phylogenetic trees.
please revise data.

Experimental design

please add raw data
Lack of accession numbers or gene identifiers to confirm phylogenetic trees.

Validity of the findings

please GenBank data

---

## Round 0.2 · accepted · Accept

The manuscript has been improved in full accordance with the comments of the reviewers.

Reviewer 2 ·

Basic reporting

Clear study written in professional English with sufficient bibliographic references. The article structure was conform.

Experimental design

The scientific approach was rigorous.

Validity of the findings

Research results were clear and well discussed.

Additional comments

The manuscript should be accepted as is.